# Differential Kinetics of Cycle Threshold Values during Admission by Symptoms among Patients with Mild COVID-19: A Prospective Cohort Study

**DOI:** 10.3390/ijerph18158181

**Published:** 2021-08-02

**Authors:** Teppei Sakano, Mitsuyoshi Urashima, Hiroyuki Takao, Kohei Takeshita, Hiroe Kobashi, Takeo Fujiwara

**Affiliations:** 1Division of Innovation for Medical Information Technology, The Jikei University School of Medicine, Tokyo 105-8461, Japan; sakano@allm.inc (T.S.); takao@jikei.ac.jp (H.T.); k.takeshita@jikei.ac.jp (K.T.); 2Allm, Inc., Yushin Bldg. Shinkan 2F, 3-27-11 Shibuya, Shibuya-ku, Tokyo 150-0002, Japan; 3Division of Molecular Epidemiology, The Jikei University School of Medicine, Tokyo 105-8461, Japan; 4Department of Global Health Promotion, Tokyo Medical and Dental University, Tokyo 113-8510, Japan; fujiwara.hlth@tmd.ac.jp; 5Department of Neurosurgery, The Jikei University School of Medicine, Tokyo 105-8461, Japan; 6Infectious Disease Department, Team Medical Clinic, Tokyo 105-0003, Japan; kobashi@team-medical.or.jp

**Keywords:** coronavirus disease 2019, COVID-19, SARS-CoV-2, mild, asymptomatic, febrile, afebrile, PCR, Ct, cycle threshold

## Abstract

In the coronavirus disease 2019 (COVID-19) pandemic, more than half of the cases of transmission may occur via asymptomatic individuals, which makes it difficult to contain. However, whether viral load in the throat during admission is different between asymptomatic and symptomatic patients is not well known. By conducting a prospective cohort study of patients with asymptomatic or mild COVID-19, cycle threshold (Ct) values of the polymerase chain reaction test for COVID-19 were examined every other day during admission. The Ct values during admission increased more steadily in symptomatic patients and febrile patients than in asymptomatic patients, with significance (*p* = 0.01 and *p* = 0.004, respectively), although the Ct values as a whole were not significantly different between the two groups. Moreover, the Ct values as a whole were higher in patients with dysosmia/dysgeusia than in those without it (*p* = 0.02), whereas they were lower in patients with a headache than those without (*p* = 0.01). Patients who were IgG-positive at discharge maintained higher Ct values, e.g., more than 35, during admission than those with IgG-negative (*p* = 0.03). Assuming that viral load and Ct values are negatively associated, the viral loads as a whole and their changes by time may be different by symptoms and immune reaction, i.e., IgG-positive at discharge.

## 1. Introduction

Currently, more than one year after the declaration of the coronavirus disease 2019 (COVID-19) pandemic by the World Health Organization (WHO) on 11 March 2020, more than 165 million people have been infected with the virus, and more than 3 million have died worldwide. Without efficient vaccinations, it seems to be difficult to control the spread of COVID-19 by only restricting human behavior. One metanalysis based on eight reports all originating from China showed that more than half of the cases of transmission may occur via presymptomatic and asymptomatic carriers [1,2], which is the most troublesome point of COVID-19 and makes it difficult to contain. In fact, presymptomatic transmission has been reported in clusters of patients with close contact approximately 1 to 3 days before the source patient developed symptoms during the early phase of the pandemic in Singapore [3] and in Germany [4]. Moreover, the possibility of asymptomatic transmission was suggested in China [5]. In contrast, another systematic review showed that the risk of transmission by asymptomatic patients was 42% lower than that by symptomatic patients [6]. Thus, infectivity of asymptomatic patients has not been well elucidated.

A lower cycle threshold (Ct) value quantified by the real-time reverse transcription-polymerase chain reaction (RT-PCR) from respiratory samples indicates large quantities of viral RNA, because the Ct values have been demonstrated to correlate strongly and negatively with cultivable virus [7]. Thus, in addition to diagnosis of COVID-19, Ct values have been used to monitor its viral load and, increasingly, infer infectivity of the patients at point-of-care testing. In fact, Ct values in the samples derived from the throat were shown to start increasing several days before the appearance of the first symptoms [8], which may support the plausibility of presymptomatic transmission. Moreover, equivalent levels of the Ct values were detected in both asymptomatic and symptomatic patients, which may suggest the plausibility of asymptomatic transmission [9,10], which may suggest the plausibility of asymptomatic transmission. However, RT-PCR positivity does not always distinguish between infectious and non-infectious virus, because RT-PCR can amplify a part of the viral genome that has already lost infectivity [11]. In fact, the median time from symptom onset to viral clearance in viral culture was 7 days, whereas the median time from symptom onset to viral clearance on RT-PCR was 34 days, and viral culture was positive only in samples with a Ct value for the N gene of 28.4 or less [12]. A Ct value for the N gene less than 40 is interpreted as positive for SARS-CoV-2 RNA in Japan.

The common symptoms of COVID-19 are fever, cough, and shortness of breath. Additional symptoms include weakness, fatigue, nausea, vomiting, diarrhea, and changes of taste and smell [1]. The severity of COVID-19 varies from asymptomatic carriers to fatal cases. Of the 104 people with SARS-CoV-2 infection on the Diamond Princess cruise ship, 32% remained asymptomatic during admission [13]. Similarly, of the 1271 crew members who were RT-PCR-positive on the U.S.S. Theodore Roosevelt Aircraft Carrier, 22% stayed asymptomatic for at least 10 weeks [14]. However, it is not well known whether viral shedding after diagnosis or during admission differs between asymptomatic and symptomatic patients, between afebrile and febrile patients, or between the presence and absence of other signs/symptoms and characteristics. Thus, the aims of this prospective cohort study enrolling asymptomatic carriers and patients with mild COVID-19 were to compare the kinetics of Ct values during admission by symptoms and other factors.

## 2. Materials and Methods

### 2.1. Ethics Statement

The protocol of this prospective cohort study was approved by the Clinical Research and Ethics Committee of the Tentakai (9-29-2020(13)), and before entry, written informed consent was obtained from all patients. This study was performed in accordance with the principles of the Helsinki Declaration.

### 2.2. Study Design

This was a prospective cohort study of asymptomatic SARS-CoV-2 carriers and patients with mild COVID-19 without hypoxia, based on the guideline by the Japanese Ministry of Health, Labour and Welfare (included if either RT-PCR or the antigen-test was positive, but excluded if the patient was age 65 years or older, had comorbidity such as diabetes mellitus or food allergy, or was pregnant), and admitted in the Shonan Village Center (SVC), located at Hayama, Kanagawa Prefecture of Japan between 17 November 2020 and 30 December 2020. SVC is a hotel, but it was a dedicated quarantine facility for COVID-19 during the study period, staffed with nurses and caregivers to support non-severe patients.

### 2.3. Participants

Consecutive patients who were positive on RT-PCR tests, i.e., Ct value < 40 at least once, with or without any symptoms, diagnosed as having mild COVID-19 or asymptomatic COVID-19, referred to and admitted to SVC, and who provided written informed consent, were enrolled during the study period.

### 2.4. MySOS

“MySOS” (Allm, Inc., Tokyo, Japan) [15] is a personal health management smartphone app that uses the communication concept of social networking services such as WhatsApp, LINE, or iMessage (Figure 1). Each participant was asked to install MySOS on their smartphone to record their daily health status and to receive consultation by healthcare professionals. “Team” (Allm, Inc., Tokyo, Japan) [16] is a web-browser based internet application, by which daily health status of each patient from MySOS was automatically gathered, monitored, and provided a consultation video call/text chat interface.

### 2.5. Cycle threshold (Ct) Values of Reverse-Transcriptase Polymerase Chain Reaction (RT-PCR) Test

RT-PCR assays for SARS-CoV-2 were performed using CronoSTAR™ 96 Real-Time PCR System (WakenBtech Co., Ltd., Kusatsu-city, Shiga, Japan) to determine the presence of the virus through the identification of nucleocapsid protein (N) gene. The Ct values of the RT-PCR test were used as indicators of the copy number of SARS-CoV-2 RNA; specimens with lower Ct values are considered to have higher viral copy numbers. According to the diagnostic criteria of the Ministry of Health, Labour and Welfare in Japan, a Ct value less than 40 was interpreted as positive for SARS-CoV-2 RNA, but a Ct value of up to 50 was recorded in this study.

### 2.6. University of Pennsylvania Smell Identification Test (UPSIT)

For the smell test, the University of Pennsylvania Smell Identification Test (UPSIT; Sensonics, Inc., Haddon Heights, NJ, USA) was administered to participants during admission. 

### 2.7. Data Monitoring during Admission

On the first day, patients were provided with the smell testing kits, saliva collection kits for the RT-PCR test, and instruction documents. They were to record, via MySOS, their name, age, sex, symptoms (cough, fatigue, shortness of breath, headache, diarrhea, and dysosmia/anosmia and/or dysgeusia/ageusia) (Figure 2), body temperature and oxygen saturation level [SpO_2_, %] using a thermometer and pulse-oximeter, respectively, medical history, drug history, and self-performed smell test and COVID-19 IgM/IgG antibody test (Artron Laboratories, Inc., Burnaby, BC, Canada) results by sending pictures of the test results.

From the second day, patients were asked to record, via MySOS, their symptoms, vitals, and self-performed smell test results every morning and every late afternoon. Saliva was collected in the late afternoon before dinner every two days during admission and also the morning of the discharge day for RT-PCR testing. Patients were discharged 10 days after symptom onset if the last 72 h had passed without symptoms. Low-grade fever was defined as follows: the temperature was slightly elevated (between 37.5 °C and 38.4 °C) and lasted for more than 24 h. Hypoxemia was defined as oxygen saturation of 93% and less.

### 2.8. Post-Discharge Questionnaire Survey

A post-discharge questionnaire survey was performed three weeks after discharge to ask about persistent symptoms in patients after acute COVID-19. PCR test-negative digital certificate for COVID-19 were provided to the participants.

### 2.9. Statistics

Skewness and kurtosis tests were used to assess the normality of the distributions. Parametric and nonparametric continuous variables with normal and nonnormal distributions were compared between two groups and among three groups using a *t*-test and Mann–Whitney test, and analysis of variance and the Kruskal–Wallis test, respectively. Dichotomous variables were compared between groups by the χ^2^ test. The kinetics of the Ct value during admission were compared between with and without a certain symptom or sign by means of repeated measure analysis of variance (ANOVA). Data were analyzed using Stata version 17.0 software (StataCorp LP, College Station, TX, USA).

## 3. Results

### 3.1. Patients’ Characteristics Stratified by Three Groups: Asymptomatic Patients, Febrile Patients, and Afebrile Patients with any Other Symptoms

A total of 121 consecutive patients agreed to participate in this study on admission to SVC. However, one patient then refused to participate. Thus, 120 patients (median age 38.9 [SD, 12.4] years; women vs. men: 51 vs. 69 years) were included in the following analyses. A total of 102 patients complained of at least one of the following symptoms on or before admission: fever, 78 (65.0%); cough, 46 (38.3%); fatigue, 40 (33.3%); headache, 22 (18.3%); sore throat, 21 (17.5%); dysfunction of smell, i.e., dysosmia, and taste, i.e., dysgeusia, 20 (16.7%); joint pain, 8 (6.7%); runny nose, 7 (5.8%); dyspnea, 7 (5.8%), and others, e.g., diarrhea, 3; dizziness, 2; appetite loss, 2; nausea/vomiting, 1; pneumonia, 1; gingivitis, 1. On the other hand, 18 people did not complain of any symptoms before and during admission, and they were defined as asymptomatic cases in this study. One of the asymptomatic carriers showed transient low-grade fever (37.9 °C), but it did not last for 24 h. Except for that, all 18 patients remained asymptomatic during admission.

The patients’ characteristics stratified as asymptomatic patients (n = 18), febrile patients (n = 78), and afebrile patients with any other symptoms (n = 24) are shown in Table 1. Age seemed to be older in febrile patients than in others, but there were no significant differences among the three groups. Duration from onset to admission was positively associated with Ct value (*p* = 0.003). Duration from admission to discharge was longer in asymptomatic patients than in febrile patients (*p* = 0.03) and afebrile patients (*p* = 0.003). The Ct values on admission of febrile patients were significantly lower than of asymptomatic patients (*p* = 0.04) and afebrile patients (*p* = 0.01). On the other hand, the minimum Ct value during admission and the Ct value at discharge were not different among the three groups. A mean of SpO2 was 96.9% (SD, 1.7%) in total, suggesting patients enrolled in this study were mild without hypoxia on admission. Results of the smell test and the IgM-positive rate on admission were not different among the three groups. No patients were IgG-positive on admission, suggesting that all participants had primary infection with COVID-19. Positivities of both IgM and IgG at discharge were also not different among the three groups. The frequencies of medical history or comorbidities, e.g., cardiac disease, respiratory disease, hypertension, and cancer, were not different among the three groups.

### 3.2. Characteristics of Three Patients Who Deteriorated after Admission and Were Referred to the Hospital

During admission, the medical condition of three patients deteriorated, and they were referred to the hospital due to hypoxia (Table 2). All three patients were male and symptomatic.

### 3.3. Differential Kinetics of Ct Values between Symptomatic and Asymptomatic Patients

First, the kinetics of Ct values during admission were compared between symptomatic (*n* = 102) and asymptomatic patients (*n* = 18) (Figure 3). The Ct values as a whole were not significantly different between asymptomatic and symptomatic patients (*p* = 0.68), whereas the Ct values increased more steadily in symptomatic patients than in asymptomatic patients with significance (*p* = 0.01). 

### 3.4. Differential Kinetics of Ct Values between Afebrile and Febrile Patients

Second, the kinetics of the Ct values during admission were compared between afebrile patients (*n* = 42) and febrile patients (*n* = 78) (Figure 4). The Ct value on the first day was lower in febrile patients than in afebrile patients (*p* = 0.01). The Ct values as a whole were not significantly different between afebrile and febrile patients (*p* = 0.54), whereas the Ct values increased more steadily in febrile patients than in afebrile patients with significance (*p* = 0.01). 

### 3.5. Differential Kinetics of Ct Values between Febrile Patients and Asymptomatic Patients

Third, by excluding afebrile patients with any symptoms other than fever (*n* = 24), the kinetics of Ct values during admission were compared between febrile patients (*n* = 78) and asymptomatic patients (*n* = 18) (Figure 5). The Ct values as a whole were not significantly different between asymptomatic and symptomatic patients (*p* = 0.91), whereas the Ct values increased more steadily in febrile patients than in asymptomatic patients with significance (*p* = 0.004). The Ct value on the first day was significantly lower in febrile cases than in asymptomatic cases (*p* = 0.04).

### 3.6. Differential Kinetics of Ct Values between Patients without and with Cough

Fourth, the kinetics of Ct values during admission were compared between patients without cough (*n* = 74) and with cough (*n* = 46) (Figure 6). There was no significant difference in the kinetics of the Ct values between the groups. However, the Ct value on admission was significantly lower in patients with cough than in those without (*p* = 0.04). 

### 3.7. Differential Kinetics of Ct Values between Patients without and with Dysosmia/Dysgeusia

Fifth, the kinetics of Ct values during admission were compared between patients without dysosmia/dysgeusia (*n* = 100) and with dysosmia/dysgeusia (*n* = 20) (Figure 7). The Ct value on the first day was higher in patients with than those without dysosmia/dysgeusia (*p* = 0.01). Similarly, the Ct values as a whole were significantly higher in patients with than those without dysosmia/dysgeusia (*p* = 0.02), although changes of the Ct values by time were not significantly different. 

There were no significant differences in the kinetics of Ct values between with and without other symptoms: fatigue, headache, and sore throat.

### 3.8. Differential Kinetics of Ct Values between Patients without and with Headache

Sixth, the kinetics of Ct values during admission were compared between patients without headache (*n* = 98) and with headache (*n* = 22) (Figure 8). The Ct values as a whole were significantly lower in patients with than those without headache (*p* = 0.01), although changes of the Ct values by time were not significantly different. 

### 3.9. Differential Kinetics of Ct Values between Patients Negative and Positive for IgM on Admission

Seventh, kinetics of Ct values during admission were compared between patients who were IgM-negative (*n* = 110) and IgM-positive (*n* = 10) on admission (Figure 9). There was no significant difference in the kinetics of Ct values between the groups, but the Ct value on the first day of admission was significantly higher in patients negative for IgM than in those positive for IgM on admission (*p* = 0.004).

IgG for SARS-CoV-2 was negative in all patients on admission.

### 3.10. Differential Kinetics of Ct Values between Patients Negative and Positive for IgG at Discharge

Eighth, the kinetics of Ct values during admission were compared between patients who were IgG-negative (*n* = 104) and IgG-positive (*n* = 12) at discharge (Figure 10). The Ct values as a whole were significantly higher in patients who were IgG-positive at discharge than in those who were IgG-negative (*p* = 0.03), although changes of the Ct values by time were not significantly different. The Ct value on the first day of admission was significantly higher in patients who were IgG-positive at discharge than in those who were IgG-negative (*p* = 0.004).

There were no significant differences in the kinetics of Ct values between those who were IgM-positive and IgM-negative at discharge.

### 3.11. Differential Kinetics of Ct Values between Women and Men

Ninth, the kinetics of Ct values during admission were compared between women (*n* = 51) and men (*n* = 69) (Figure 11). There was no significant difference in the kinetics between women and men. However, the Ct value on day 7 of admission was significantly less in men than in women (*p* = 0.03).

### 3.12. Differential Kinetics of Ct Values between Younger and Older Than Forty Years

Tenth, the kinetics of Ct values during admission were compared between younger (≤40 years of age: *n* = 60) and older (>40 years: *n* = 60) cases (Figure 12). Recovery speed, i.e., increment in Ct value, seemed faster in younger than in older cases, although there was no significant difference. In the group of younger patients, the Ct value exceeded 40 before day 7, whereas in the group of older patients, the Ct value exceeded 40 after day 8.

### 3.13. Persistent Symptoms in Patients after Acute COVID-19

A total of 96 (80%) patients replied to the post-discharge questionnaire survey three weeks after discharge. A total of 34 (35.4%) patients complained of at least one of following symptoms: dysosmia/dysgeusia, 18 (19.0%); difficulty in breathing, 13 (13.7%); cough/sputum, 12 (12.6%); fatigue, 8 (8.4%); headache, 3 (3.2%); fever, 2 (2.1%); sleepiness, 2 (2.1%); chest pain, 1 (1.1%); joint pain, 1 (1.1%). On the other hand, 62 (64.6%) people did not complain of any symptoms. There were no significant differences in the rates of any other symptoms among the three groups: asymptomatic patients, febrile patients, and afebrile patients with any other symptoms (Table 3).

## 4. Discussion

In this study, initially, the differential kinetics of Ct values during admission by symptoms were shown. The Ct values on admission were higher in asymptomatic patients than in febrile patients. Moreover, the Ct values during admission increased more steadily in symptomatic patients and febrile patients than in asymptomatic patients, although the Ct values as a whole were not significantly different between the two groups. Le Bert et al. demonstrated that SARS-CoV-2-specific T cells derived from asymptomatic patients produced higher levels of interferon gamma and interleukin 2 than those from symptomatic patients, whereas such T cells from severe patients secreted more pro-inflammatory cytokines, such as interleukin-6, than those from asymptomatic patients, suggesting that asymptomatic patients are not characterized by weak antiviral immunity, but by a highly functional, virus-specific cellular immune response [17]. In fact, IgM was positive in 16.7% of asymptomatic patients compared to 6.4% in febrile patients on admission, although the difference was not significant. Carsetti et al. showed that an early increase of specific IgM was associated with asymptomatic SARS-CoV-2 infection [18], which is consistent with the results of the present study. Thus, asymptomatic patients may suppress viral proliferation without signs of inflammation, such as fever. Assuming low Ct values are associated with more viral load or being more infectious, asymptomatic patients may be less infectious than febrile or patients with symptoms such as coughing and headache, because the means of the Ct values in asymptomatic patients were always above 35 during admission, whereas the means in febrile or symptomatic patients were lower than 35 in the first couple of days after admission. This hypothesis is consistent with the systematic review showing that the risk of transmission by asymptomatic patients was 42% lower than that by symptomatic patients [6]. The Ct values on admission were higher in patients with dysosmia/dysgeusia than in those without it. Alterations in smell and/or taste were frequently reported by mildly symptomatic patients with SARS-CoV-2 infection and were often the first apparent symptom [19,20]. On the other hand, the Ct values as a whole were lower in patients with than those without headache. However, why the Ct values were higher in patients with dysosmia/dysgeusia and lower in patients with headache remains unknown.

Next, the differential kinetics of Ct values during admission by positivity of IgM or IgG either on admission or at discharge were examined. Patients who were IgM-positive on admission and IgG-positive at discharge had already high Ct values, e.g., more than 35, but patients who were IgM- or IgG-negative had low Ct values on admission. In addition, patients who were IgG-positive at discharge maintained higher Ct values, e.g., more than 35, during admission than those with IgG-negative. These results suggest that IgM- or IgG-positive patients may be less infectious than IgM- or IgG-negative patients.

Finally, the kinetics of Ct values were compared between women and men and between young and old persons. The Ct values increased more quickly in female patients than in male patients, indicating that female patients may recover faster than male patients. Indeed, the immune landscape in patients with COVID-19 is considerably different between the sexes, and these differences may underlie heightened disease vulnerability in men [21,22]. Thus, these immunological differences between women and men were thought to affect the kinetics of Ct values. Similarly, Ct values seemed to recover faster in young than in old persons, although no significant difference was detected.

There are several limitations to this study. First, Ct values were used as a parameter reflecting viral load in this study, because a previous study demonstrated that Ct values correlate strongly with cultivable virus [7]. However, RT-PCR positivity does not always distinguish between infectious and non-infectious virus, because RT-PCR can amplify a part of the viral genome that has already lost infectivity [23]. Second, the sample size was too small to detect significant differences in some comparisons. Third, analyses of symptoms and other factors may increase the probability of type I error due to multiple comparisons. Thus, some of the significant results may be due to chance. Fourth, because this study was conducted in Japan, the results of the present study are not necessarily generalizable to other populations or countries. Fifth, high Ct values, i.e., around 30, were observed on admission in this study, compared with other published studies [11,24]. Rao et al. suggested that higher Ct values may be associated with better outcome [11]. Patients analyzed in this study were asymptomatic carriers or mild cases without hypoxia. That is why the overall Ct values were high in this study. Moreover, N gene, but not COVID-19-RdRp/Hel [25] nor S gene [24], was used for PCR testing. Duration from onset to admission was median 4 to 5 days. In this study, these factors may also raise Ct value on admission. Sixth, changes of Ct values by saliva PCR test during admission were applied in this study. However, a recent Cochrane systematic review [26] has found conflicting reports on this approach between saliva testing and nasopharyngeal testing as a golden standard. Seventh, days from onset to admission were different among patients, which may be a potential source of bias on Ct values on admission. Eighth, regarding persistent symptoms, a total of 35.4% patients complained of at least one symptom. However, 80% replied to the post-discharge questionnaire survey; thus, this percentage may include bias. Ninth, scores of the smell test on admission used in this study as an objective measure did not always match dysosmia as a subjective symptom, although a study using UPSIT to compare smell acuity in patients diagnosed with COVID-19, compared with a matched control group, showed that 98% of the patient group exhibited some smell dysfunction, scoring significantly lower on the UPSIT compared to controls [27].

## 5. Conclusions

Assuming that viral load and Ct values are associated, the results of this study generated the hypothesis that the viral load may be (1) decreased more steadily in symptomatic patients and febrile patients than in asymptomatic patients, (2) low in patients with dysosmia/dysgeusia and high in patients with headache, and (3) low in patients who were IgG-positive at discharge and high in those who were IgG-negative.

## Figures and Tables

**Figure 1 ijerph-18-08181-f001:**
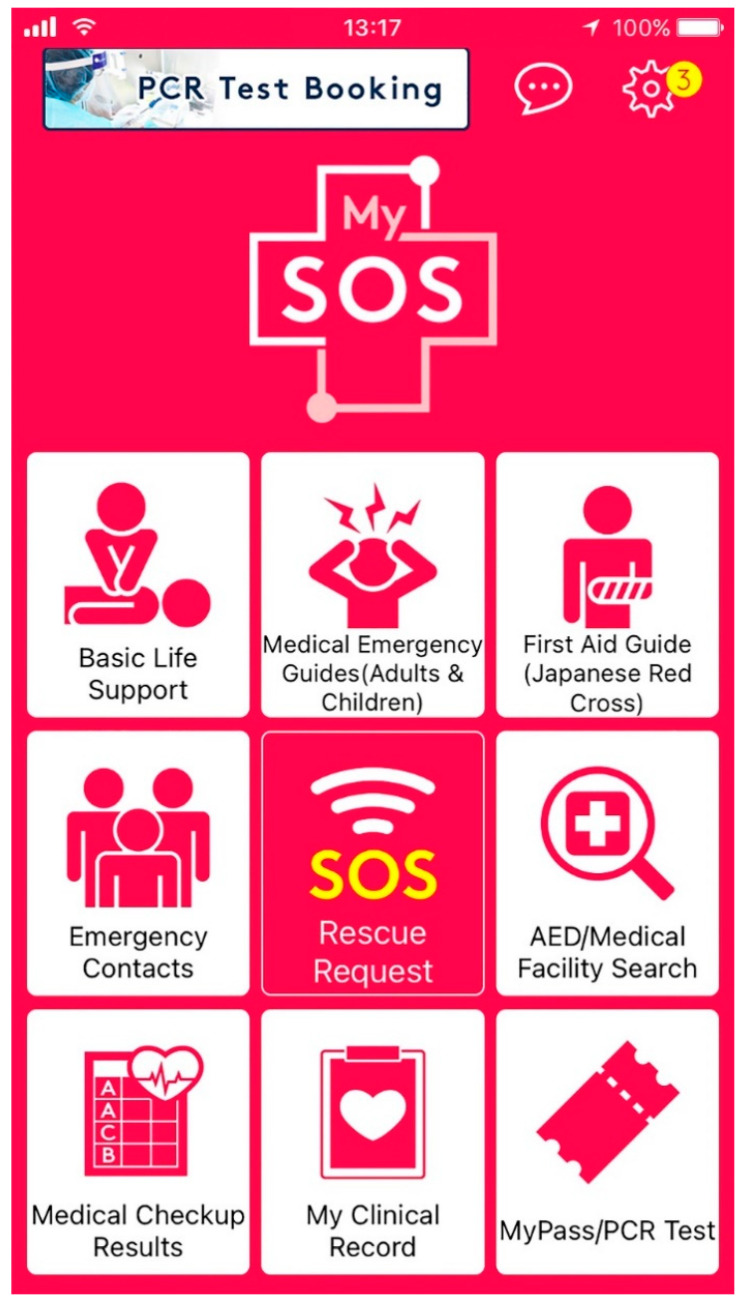
MySOS interface.

**Figure 2 ijerph-18-08181-f002:**
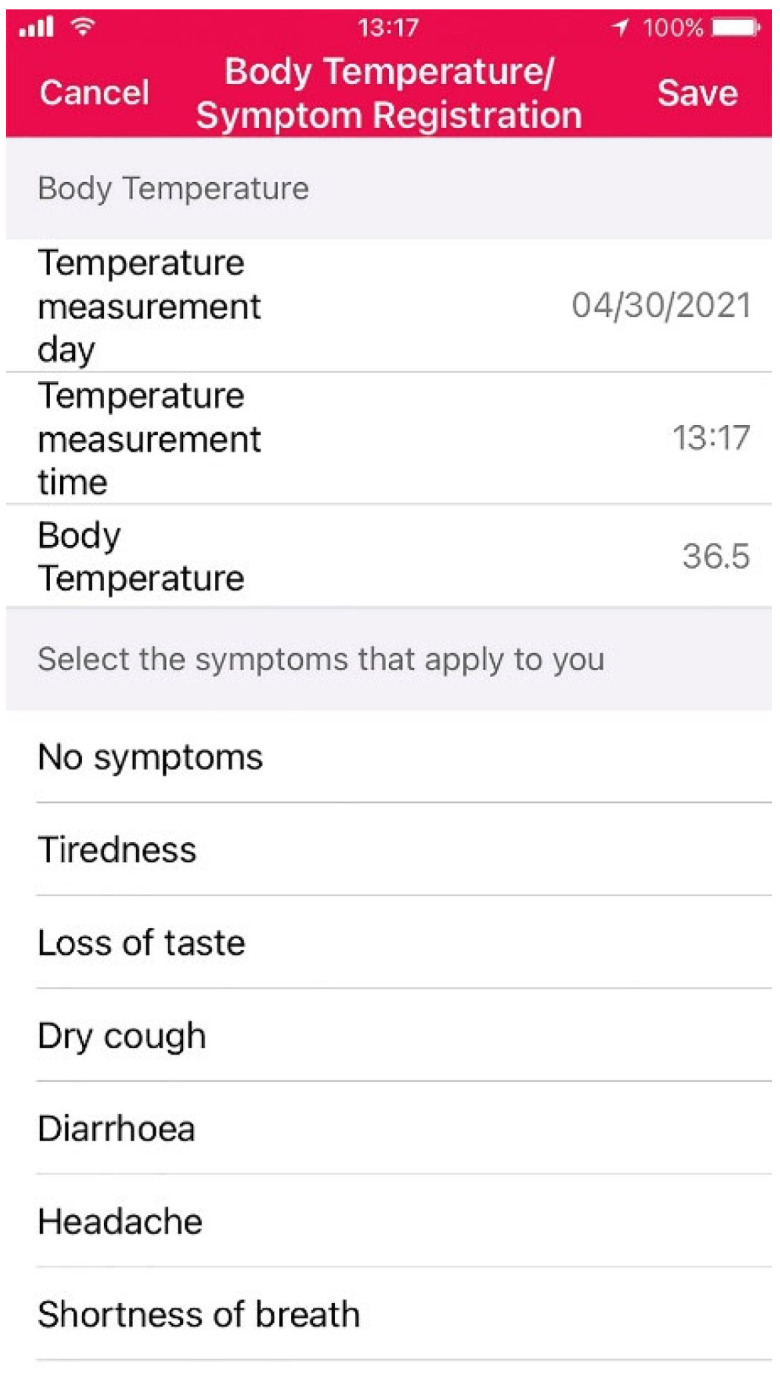
Sample of body temperature/symptoms. Users can register and manage their daily health condition and symptoms as screening items for COVID-19.

**Figure 3 ijerph-18-08181-f003:**
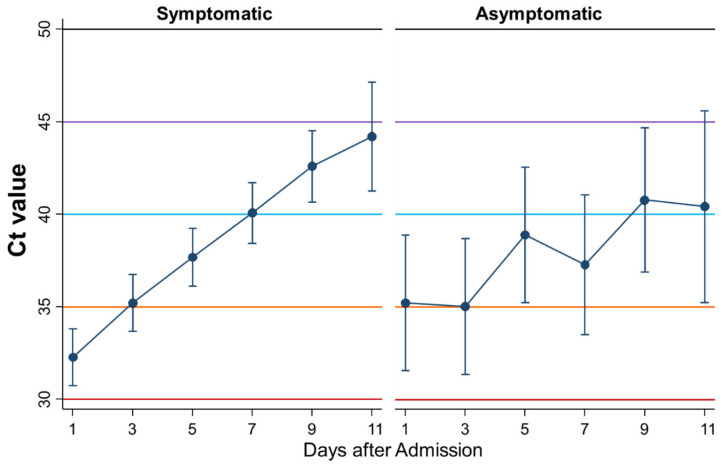
Differential kinetics of Ct values between symptomatic and asymptomatic patients. Mean Ct with 95% confidence intervals. When the Ct value is 40 or more, the PCR test result is considered negative.

**Figure 4 ijerph-18-08181-f004:**
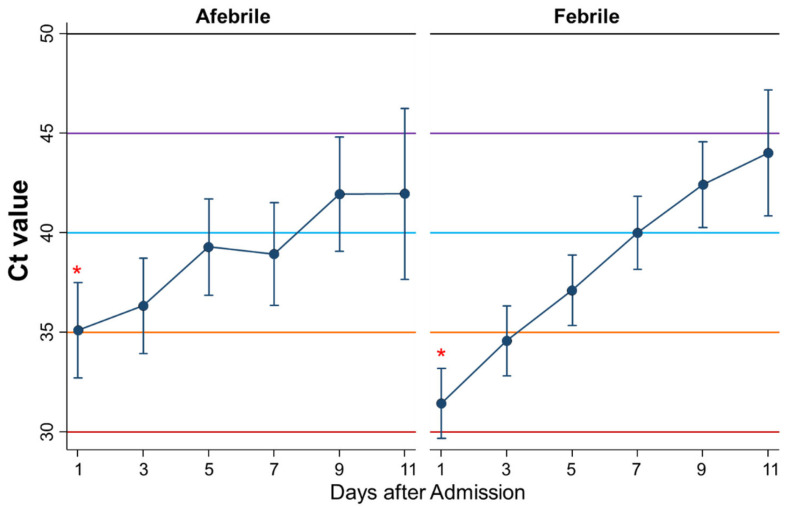
Differential kinetics of the Ct values between afebrile and febrile patients. Mean Ct with 95% confidence intervals. When the Ct value is 40 or more, the PCR test result is considered negative. * represents a significant difference (*p < 0.05*).

**Figure 5 ijerph-18-08181-f005:**
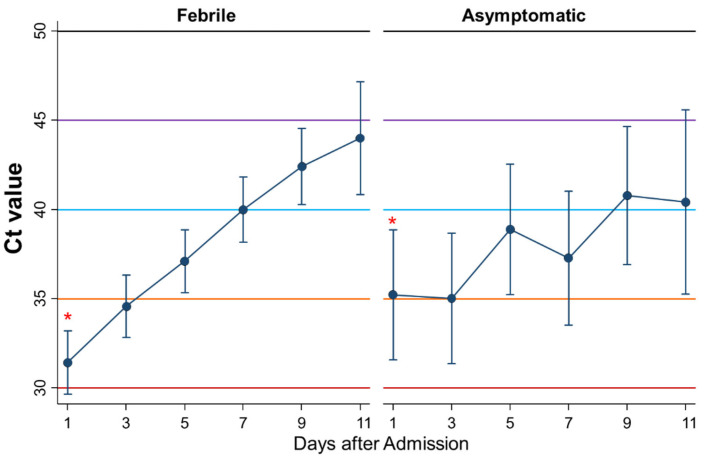
Differential changes of Ct values between febrile and asymptomatic patients. Mean Ct with 95% confidence intervals. When the Ct value is 40 or more, the PCR-test result is considered negative. * represents a significant difference (*p < 0.05*).

**Figure 6 ijerph-18-08181-f006:**
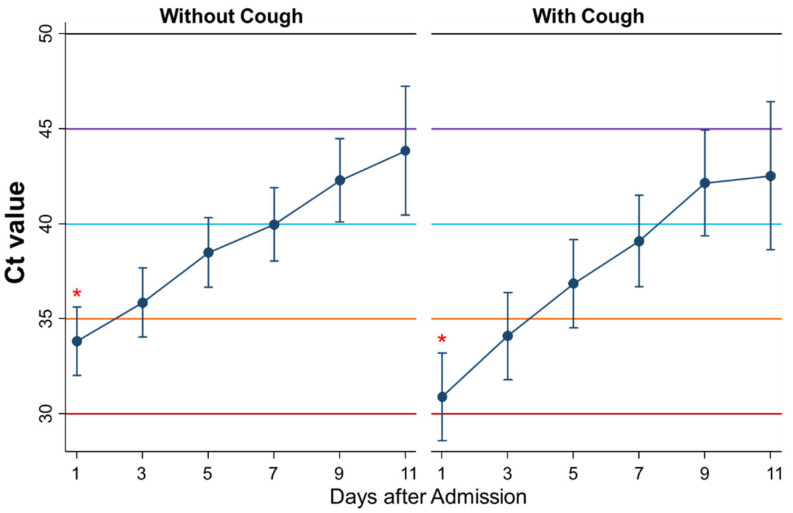
Differential kinetics of Ct values between patients without and with cough. Mean Ct with 95% confidence intervals. When the Ct value is 40 or more, the PCR test result is considered negative. * represents a significant difference (*p* < 0.05).

**Figure 7 ijerph-18-08181-f007:**
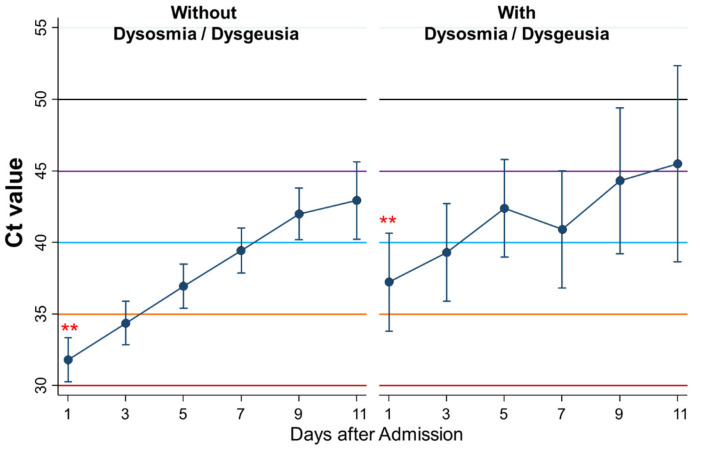
Differential kinetics of Ct values between patients without dysosmia/dysgeusia and with dysosmia/dysgeusia. Mean Ct with 95% confidence intervals. When the Ct value is 40 or more, the PCR test result is considered negative. ** represents a significant difference (*p* < 0.005).

**Figure 8 ijerph-18-08181-f008:**
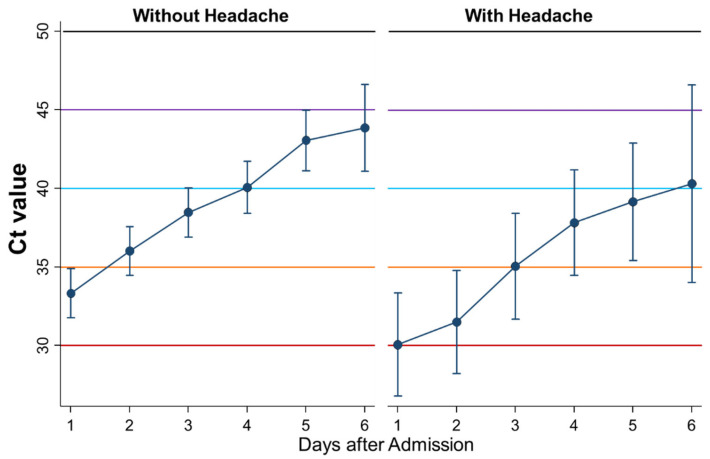
Differential kinetics of Ct values between patients without headache and with headache. Mean Ct with 95% confidence intervals. When the Ct value is 40 or more, the PCR test result is considered negative.

**Figure 9 ijerph-18-08181-f009:**
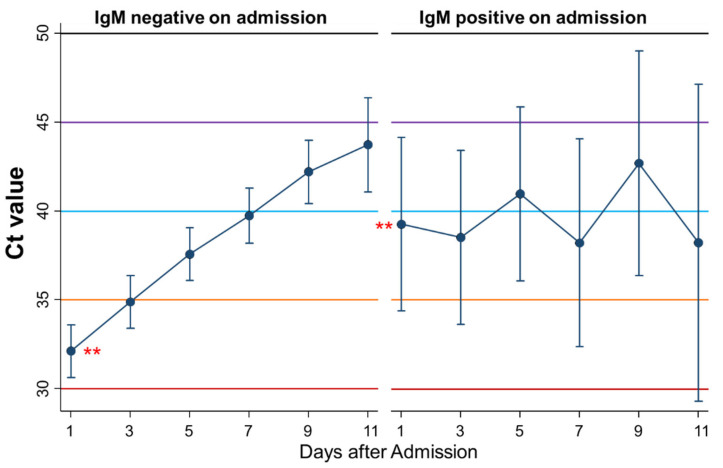
Differential kinetics of Ct values between patients who were IgM negative and IgM positive on admission. Mean Ct with 95% confidence intervals. When the Ct value is 40 or more, the PCR test result is considered negative. ** represents a significant difference (*p* < 0.005).

**Figure 10 ijerph-18-08181-f010:**
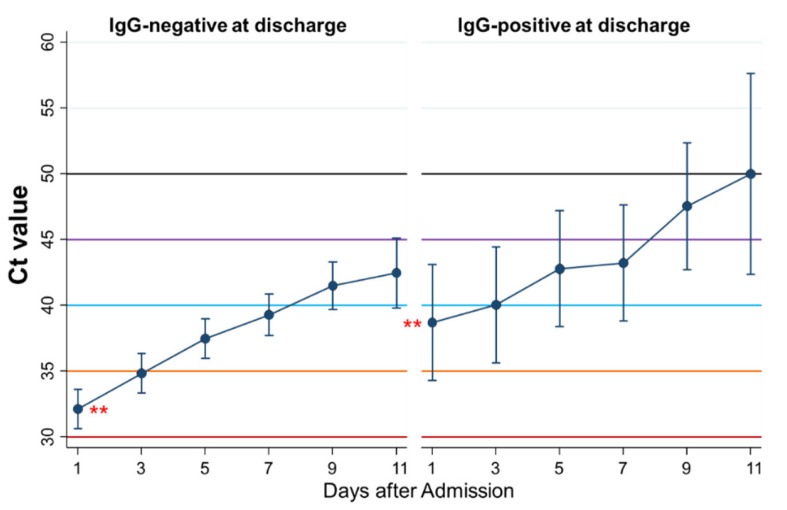
Differential kinetics of Ct values between patients with negative and positive IgG at discharge. Mean Ct with 95% confidence intervals. When the Ct value is 40 or more, then the PCR test result is considered negative. ** represents a significant difference (*p* < 0.005).

**Figure 11 ijerph-18-08181-f011:**
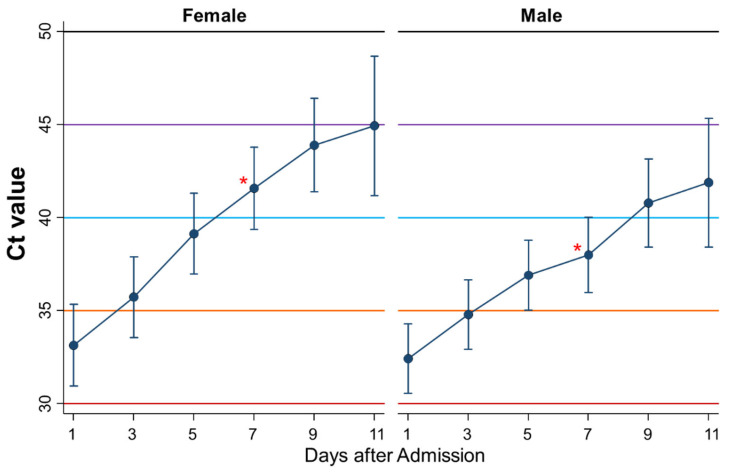
Differential kinetics of Ct values between female and male patients. Mean Ct with 95% confidence intervals. When the Ct value is 40 or more, the PCR test result is considered negative. * represents a significant difference (*p* < 0.05).

**Figure 12 ijerph-18-08181-f012:**
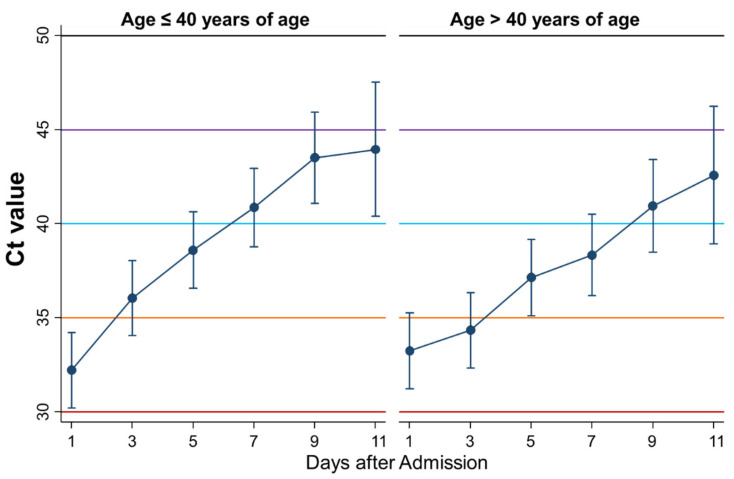
Differential kinetics of Ct values between age ≤ 40 y and >40 y. Mean Ct with 95% confidence intervals. When the Ct value is 40 or more, then the PCR test result is considered negative.

**Table 1 ijerph-18-08181-t001:** Patients’ characteristics stratified by three groups: asymptomatic, dysosmia/dysgeusia, and others.

Variable	Asymptomatic *n* = 18	Symptomatic *n* = 102		*p*-Value
		Febrile *n* = 78	Afebrile *n* = 24	
Age, y Median (IQR ^1^)	37 (24–48)	43.5 (29–50)	38.5 (26–47)	0.72 ^4^
Men, n (%)	12 (66.7%)	44 (56.4%)	13 (54.2%)	0.68 ^5^
Onset to admission, days Median (IQR ^1^)	-	4 (3–5)	5 (3–6.5)	0.16 ^6^
Admission to discharge, days Median (IQR ^1^)	10 (9–10)	9 (7–10)	8 (6–9)	0.01 ^4^
Onset to discharge, days Median (IQR ^1^)	-	12 (12–12)	12 (12–12)	0.15 ^6^
Ct value on day 1 Mean (SD ^1^)	32.7 (28.5–39.7)	30.1 (26.9–35.2)	34.4 (28.4–38.8)	0.04 ^7^
Minimum Ct value Mean (SD ^1^)	30.3 (25.9–35.7)	29.1 (26.0–31.6)	31.8 (27.1–36.8)	0.14 ^7^
Ct value on discharge Mean (SD ^1^)	39.0 (35.2–50)	41.9 (34.7–50)	47.1 (36.4–50)	0.76 ^7^
UPSIT for smell test ^3^				0.29 ^5^
0~24%	5 (27.8%)	14 (18.0%)	9 (40.9%)	
25~49%	2 (11.1%)	5 (6.4%)	3 (13.6%)	
50~74%	1 (5.7%)	14 (18.0%)	3 (13.6%)	
75~99%	3 (16.7%)	20 (25.6%)	3 (13.6%)	
100%	7 (38.9%)	25 (32.1%)	4 (18.2%)	
SpO_2_, % ^2^ Median (IQR ^1^)	97 (97–97)	97 (97–98)	97 (96–97)	0.20 ^4^
IgM on admission				0.37 ^5^
-	15 (83.3%)	73 (93.6%)	22 (91.7%)	
±	0 (0.0%)	0 (0.0%)	0 (0.0%)	
+	3 (16.7%)	5 (6.4%)	2 (8.3%)	
IgG on admission				-
-	18 (100%)	78 (100%)	24 (100%)	
±	0 (0.0%)	0 (0.0%)	0 (0.0%)	
+	0 (0.0%)	0 (0.0%)	0 (0.0%)	
IgM at discharge				0.96 ^5^
-	7 (38.9%)	25 (32.9%)	8 (34.8%)	
±	3 (16.7%)	19 (25.0%)	5 (21.7%)	
+	8 (44.4%)	32 (42.1%)	10 (43.5%)	
IgG at discharge				0.44 ^5^
-	16 (88.9%)	63 (82.9%)	18 (78.3%)	
±	0 (0.0%)	7 (9.2%)	1 (4.4%)	
+	2 (11.1%)	6 (7.9%)	4 (17.4%)	

^1^ IQR: interquartile range (25–75%), SD: standard deviation ^2^ The questionnaire was collected from 96 participants. ^3^ The data were obtained on the first day of admission. ^4^
*p*-value calculated by the Kruskal–Wallis equality-of-populations rank test. ^5^
*p*-value calculated by the chi-squared test. ^6^
*p*-value calculated by the Mann–Whitney test. ^7^
*p*-value calculated by analysis of variance.

**Table 2 ijerph-18-08181-t002:** Characteristics of patients referred to the hospital.

Patient	Age (y) Sex	Onset Date Admission Date	Symptoms	Ct Value on Admission	Minimum O_2_ Saturation	Date of Referral	Reason for Referral
1	39 male	11/11/2020 13/11/2020	Fever Dysosmia	28.2	93	18/11/2020	hypoxia
2	54 male	18/11/2020 21/11/2020	Cough	34.3	82	23/11/2020	hypoxia
3	58 male	1/12/2020 6/12/2020	Fever, Cough, Sore throat	24.8	93	18/11/2020	hypoxia

**Table 3 ijerph-18-08181-t003:** Post-discharge questionnaire survey stratified by three groups.

Variable *n* (%) ^1^	Asymptomatic *n* = 15	Symptomatic *n* = 102		*p*-Value ^2^
		Febrile *n* = 64	Afebrile *n* = 17	
Dysosmia/dysgeusia	2 (13.3%)	12 (19.1%)	4 (23.5%)	0.76
Difficulty in breath	1 (6.7%)	10 (15.9%)	2 (11.8%)	0.63
Cough/sputum	1 (6.7%)	6 (9.5%)	5 (29.4%)	0.07
Fatigue	0 (0.0%)	5 (7.9%)	3 (17.7%)	0.19
Headache	0 (0.0%)	2 (3.2%)	1 (5.9%)	0.64
Fever	0 (0.0%)	2 (3.2%)	0 (0.0%)	0.60
Sleepiness	1 (6.7%)	0 (0.0%)	1 (5.9%)	0.13
Chest pain	0 (0.0%)	1 (1.6%)	0 (0.0%)	0.77
Joint pain	0 (0.0%)	1 (1.6%)	0 (0.0%)	0.77
Any chronic symptoms, n (%) ^2^	4 (26.7%)	23 (36.5%)	7 (41.2%)	0.73

^1^ The questionnaire was collected from 96 participants. ^2^
*p*-value was calculated by the chi-squared test.

## Data Availability

The data presented in this study are available on request from the corresponding author.

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
