# Peer review of "Differential Kinetics of Cycle Threshold Values during Admission by Symptoms among Patients with Mild COVID-19: A Prospective Cohort Study"

_ijerph, 2021, doi:10.3390/ijerph18158181_

Round 1
Reviewer 1 Report
A very relevant, well conducted and well presented study; I think that just two aspects are worthy of further information and discussion.
i) Which viral component was choosen in this study to conduct PCR testing (S, N, RdRp, E, other) ?
ii) The Authors found an overall pattern of high Ct values (standing for low viral loads) at the very beginning of the infection too, with a minimum mean value of 30.1 (SD 26.9 - 35.23) in the subgroup of their febrile patients (Table 1, row "Ct value on day 1") and Ct values of respectively 28.2, 34.3 and 24.8 on admission for their three patients requiring hospitalization (Table 2, column 5). Other published studies signalled definitely lower Ct (see e.g. Rao et al 2020 and Ade et al 2021). In Italy Ct values for the N gene low up to 13 - 15 were not exceptionally detected (unpublished data). I modestly propose that an analysis of the possible determinants of these differences (kind of viral component choosen for Ct testing ? differences in the typical national / local habits of interpersonal relationships, conditioning the doses - intensity and / or duration - of the exposures to SARS-CoV-2 ? others ?) will be of great interest for Public Health decision making.

Reviewer 2 Report
The manuscript attempts to correlate clinical data collected from asymptomatic patients and patients with mild COVID-19 with RT-PCR Cts as a surrogate for SARS-CoV-2 viral loads and, through extrapolation, infectivity. There are some important limits to the study as described: a major one is that no proper information is provided about the characteristics of the PCR test (tests?) that have been used. There are broad differences between tests which make comparison of Cts problematic. Even for a single test and within a single laboratory, there are significant differences among different targets: the N gene, for instance, is well known for being detectable long periods of time after the symptoms subsided. The authors do not inform us about the PCR target (targets?) the Cts are measured for. Not least, there are broad differences among reports when linking particular Ct thresholds and infectivity; ideally, such a threshold should be described.
The authors use saliva as a biological product to be tested by RT-PCR. A recent Cochrane systematic review (Dinnes et al. 2021) has found conflicting reports on this approach. Saliva testing should not be used unless it had been evaluated against naso-faringial testing as a golden standard.
The monitoring of the symptoms is recorded starting from the day of admission; this is a potential source of bias, since the onset might have preceded hospitalization by different intervals and the durations of symptoms (and viral loads) might as well differ accordingly.
The distinction between presymptomatic, asymptomatic and, possibly, postsymptomatic patients is not discussed.
Some of the figures (e.g. 3, 4, 5) have little – if any – relevance to the study.
Specific comments:
Lines 38-39: More than half of the cases 38 of transmission may occur via presymptomatic and asymptomatic carriers [1]: the reference is one metanalysis based on 8 reports all originating from China; the authors should put in balance the role of superspreaders and the largely accepted Pareto principle.
Line 45: true asymptomatic infection is probably uncommon [5],: the metaanalysis in reference [5] finds that the transmission risk from asymptomatic persons is 42% lower thus contradicting the statement in lines 38-39.
Lines 54-56: Moreover, equivalent levels of the Ct values were detected in both asymptomatic and symptomatic patients, which may suggest the plausibility of asymptomatic transmission [8].: This is an early study with a small number of patients; several other studies suggest otherwise.
Line 61: viral culture was positive only in samples with a Ct value of 28.4 or less [10].: the target gene should be specified every time since the dynamic of Cts for different genes may differ in the same patient; in the cited article, it is the Ct for N gene - which is known to be detectable for longer periods.
Lines 95-127: the extensive description of the app is of very low relevance to the study.
Lines 128-133: the RT-PCR test, the target gene(s) and the laboratory methodology should be thoroughly described.
Lines 137-139: A study using UPSIT to compare smell acuity in patients diagnosed with COVID-19, com-137 pared with a matched control group, showed that 98% of the patient group exhibited some 138 smell dysfunction, scoring significantly lower on the UPSIT compared to controls [15].: statement rather fit for the discussion section.
Lines 370-371: Assuming low Ct values, 370 e.g., less than 35, are associated with more viral load or being more infectious… : 35 is an arbitrary value and may have very different significance with different tests and gene targets.
Round 2
Reviewer 2 Report
The authors have responded in a satisfactory manner to most of the observations formulated during the first review. Nevertheless, a major problem persists: it turns out that the Ct measurements have been performed with two different assays; this might have a significant effect on the analysis. Either the two sets of data should be analysed separately or a single more substatial set of data generated with a single test should be presented alone. Another solution would be the authors to perform validation experiments that show the dynamics of the gene N detection to be similar for the two assays.
